# Lignosulfonate-Based Ionic Liquids as Asphaltene Dispersants

**DOI:** 10.3390/molecules28083390

**Published:** 2023-04-12

**Authors:** Ariff Mahtar, Aliyu Adebayo Sulaimon, Cecilia Devi Wilfred

**Affiliations:** 1Centre of Research in Ionic Liquids, Universiti Teknologi Petronas, Bandar Seri Iskandar 32610, Malaysia; 2Department of Petroleum Engineering, Universiti Teknologi Petronas, Bandar Seri Iskandar 32610, Malaysia; 3Fundamental and Applied Sciences Department, Universiti Teknologi Petronas, Bandar Seri Iskandar 32610, Malaysia

**Keywords:** ionic liquids, lignosulfonate, piperidinium, asphaltene, dispersant

## Abstract

Asphaltenes are recognized as being troublesome from upstream to downstream in the oil industry due to their tendency to precipitate and self-associate. Their extraction from asphaltenic crude oil for a cost-effective refining process is a crucial and critical challenge in the oil and gas sector. Lignosulfonate (LS), as a by-product of the wood pulping process in the papermaking industry, is a highly available and underutilized feedstock. This study aimed to synthesize novel LS-based ionic liquids (ILs) by reacting lignosulfonate acid sodium salt [Na]_2_[LS] with different alkyl chains of piperidinium chloride for asphaltene dispersion. The synthesized ILs, 1-hexyl-1-methyl-piperidinium lignosulfonate [C_6_C_1_Pip]_2_[LS], 1-octyl-1-methyl-piperidinium lignosulfonate [C_8_C_1_Pip]_2_[LS], 1-dodecyl-1-methyl-piperidinium lignosulfonate [C_12_C_1_Pip]_2_[LS] and 1-hexadecyl-1-methyl-piperidinium lignosulfonate [C_16_C_1_Pip]_2_[LS] were characterized using FTIR-ATR and ^1^H NMR for functional groups and structural confirmation. The ILs depicted high thermal stability because of the presence of a long side alkyl chain and piperidinium cation following thermogravimetric analysis (TGA). Asphaltene dispersion indices (%) of ILs were tested by varying contact time, temperature and ILs concentration. The obtained indices were high for all ILs, with a dispersion index of more than 91.2% [C_16_C_1_Pip]_2_[LS], representing the highest dispersion at 50,000 ppm. It was able to lower asphaltene particle size diameter from 51 nm to 11 nm. The kinetic data of [C_16_C_1_Pip]_2_[LS] were consistent with the pseudo-second-order kinetic model. The dispersion index (%), asphaltene particle growth and the kinetic model agreed with the molecular modeling studies of the HOMO–LUMO energy of IL holds.

## 1. Introduction

Asphaltene has an exceptionally high molecular weight and is the largest component of crude oil [1]. Asphaltene is also known as petroleum fume, which is soluble in aromatic compounds and insoluble in *n*-alkanes [2]. It is a suspended/homogenized solid phase in crude oil at reservoir conditions [3]. Asphaltene is one of the highly polar components in crude oil as opposed to the non-polar media of crude oil [2]. Asphaltene crowding can form “skin” at the interface which is influenced by self-association at the fluid interface, consequently initiating the formation of layers. These layers turn into high-elasticity films after many hours of ageing [3]. Thus, petroleum operations and processes face thick asphaltene flocculation formation and deposit issues in reservoirs, wellbores and transportation pipes. Asphaltene agglomerates can occur at nanoaggregate concentrations in the order of 100 mg/L [4]. The creation of the asphaltene aggregation nanoaggregate leads to the formation of asphaltene molecules, which eventually form clusters [5]. Conventionally, steam and carbon dioxide gas injection or diluent addition alter its rheological properties. However, these additives gradually affect the reservoir porosity and downstream pressure vessels [6]. Therefore, asphaltene extraction from asphaltenic crude oil for a cost-effective refining process is a crucial and critical challenge in the oil and gas sector.

Conventionally, the usage of a variety of organic solvents with different solubility parameters can enhance the stability of asphaltene in oil [7]. Blended aromatic solvents of xylene and toluene have been used to dissolve and cast off asphaltic and asphaltene deposits, respectively, from an oil pipeline. However, volatile organic compounds (VOC) solvents have been reported as carcinogenic and harmful to the environment [8]. Alkyl benzene sulfonic acid and amine as alternative asphaltenes solvents have enhanced the inhibition of asphaltene precipitation via acid–base interaction [9]. The interaction of asphaltene and amphiphile forms the irreversible bond between them because the electrophilic addition reaction takes place [9]. The direct adsorption of the amphiphile solvents at the surface of asphaltene molecules is eventually responsible for the formation of large conjugated systems. The solubility of asphaltene diminishes after the solvents associated with asphaltene in non-polar media form a precipitate [10].

Ionic liquids (ILs) are another class of solvents which have been employed to aid in asphaltene aggregation problems due to their several unique properties, such as high chemical stability, negligible volatility, low melting point and high thermal stability [11,12]. The thermal stability of ILs is affected by anion type [13], cation type [14], allyl-functionalization [15] and anion-coordinating nature [16]. The vapor pressure of ILs remains negligible at near ambient temperature [17] and ILs show no sign of distillation below the temperature of their thermal decomposition [18]. The investigations into ILs have been widened for various applications, for example in heterogeneous catalysts [19], biomedicine [20], lithium-ion batteries [21], extraction from plant biomass [22], sulfur and nitrogen removal from fuels [23], dopants [24], solvents for organic reaction at high temperature [25], and biomass dissolution using amino acid ILs (AAILs) [26] and acetate-based ILs (Ac-ILs) [27]. The careful selection of cations and anions has been carried out to tailor targeted properties for asphaltene stability by recognizing the interaction and mechanism between ILs and asphaltenes. The details and factors of the ILs used that influence the asphaltene dispersion and aggregation are compiled in Table 1.

ILs are suggested to incorporate with cation heads that have a high non-polarity property with charge-sharing ability which implies better interaction with the heteroatoms of asphaltene molecules. In addition, the longer side alkyl chain of the cation head can exhibit non-polarity, less compactness and a high-affinity environment of IL towards asphaltene dissolution. Functional groups of anions can alter the reactive ability of an anion. The presence of a phenyl group in an anion induces hydrophobicity due to the Van Der Waals interaction towards asphaltene molecules. The high polarity of anions, contributed by the presence of the heteroatoms of fluoride (F), oxygen (O) and nitrogen (N) in the anion, can consequently enhance the dispersing strength of ILs towards asphaltene molecules via a hydrogen bond interaction towards the heteroatoms of the molecules.

Biomass valorization and transformation processes convert waste materials into value-added products such as ionic liquids [17,32,33], bioproducts [34] and enzymes [35]. Lignin is one of the largest biomasses in terrestrial plants [36]. Technical lignin refers to the derivative obtained as the product of the delignification process of lignocellulosic-biomass [37], such as lignosulfonate lignin, kraft lignin, organosolv lignin and soda lignin. The lignosulfonate structure can be varied through chemical reactions involved in biomass treatment. Conventionally, lignosulfonate has been produced as a by-product during the pulping process. An aqueous solution containing sulfurous acid salt and sulfur dioxide is responsible for breaking down polymer linkages into polysaccharides [38]. Gu and Bai et al. have modified the polyelectrolyte sodium lignosulfonate [Na]_2_[LS] via a cation exchange reaction using a sodium base [39]. The recent technology of lignin-centric biorefinery has welcomed the by-product lignosulfonate’s treatment strategies, thus providing an impetus and interest for application studies [40]. The recent applications of lignosulfonate are compiled in Table 2.

Nevertheless, less attention has been given to the development and mechanism of lignosulfonate-based ILs to be utilized in inhibiting asphaltene aggregation. Lignosulfonate as a dianion consists of hydrophobic moieties such as aromatic rings and aliphatic linkages, and hydrophobic moieties such as sulfonate and hydroxyl groups, which account for its amphiphilic property [46]. Thus, lignosulfonate-based ILs may have possible π–π and hydrogen bonding interactions towards asphaltene molecules since the molecules consist of polynuclear aromatic rings and heteroatoms of nitrogen (N), oxygen (O) and sulfur (S).

The piperidinium cation was selected as it was reported that the aliphatic ring showed a higher affinity towards asphaltene molecules compared to aromatic rings due to the delocalization of charge in the aromatic ring [22].

To our knowledge, there has been no reported work on employing piperidinium lignosulfonate for asphaltene dispersion. The purpose of this work is to synthesize a series of piperidinium lignosulfonate ionic liquids comprising the following: 1-hexyl-1-methyl-piperidinium lignosulfonate [C_6_C_1_Pip]_2_[LS], 1-octyl-1-methyl-piperidinium lignosulfonate [C_8_C_1_Pip]_2_[LS], 1-dodecyl-1-methyl-piperidinium lignosulfonate [C_12_C_1_Pip]_2_[LS] and 1-hexadecyl-1-methyl-piperidinium lignosulfonate [C_16_C_1_Pip]_2_[LS]. The piperidinium cation has been selected for this study and its side alkyl chain (*n* = 6, 8, 12 and 16) has been varied via quaternization. Synthesized ILs were characterized for functional groups, structural confirmation and thermal stability studies. The synthesized ILs were tested on model oil to investigate the asphaltene dispersion index. The IL with the best dispersion index was optimized using both particle size analysis study and kinetic study. Pseudo-first-order and second-order kinetic models were applied to predict the asphaltene extraction phenomenon involved. Molecular modeling of the best performance index IL was employed to determine local forces acting within the selected individual IL, and to determine possible interactions between employed IL and asphaltene models.

## 2. Results and Discussion

### 2.1. FTIR

Synthesized ILs were characterized using FTIR for functional group confirmation. The spectrum of the ILs is shown in Figure 1 and summarized in Table 3. The O-H and N-H stretches appeared at 3350 cm^−1^ [32,33]. The C-H stretch in the aliphatic chain appeared at 2930 cm^−1^. The peak at 1560 cm^−1^ is the unsaturated C=C stretch in the aromatic ring. The C-H bend in polynuclear aromatic rings appeared at 1460 cm^−1^. C-N stretch peaks were at 1400 cm^−1^ [47]. There was a CH_3_ bend peak at 1350 cm^−1^. S=O stretch peaks appeared at approximately 1110 cm^−1^. The peak at 1050 cm^−1^ is attributed to the C-O stretching of methoxy groups.

### 2.2. ^1^H NMR

The synthesized ionic liquids [C_6_C_1_Pip][Cl], [C_8_C_1_Pip][Cl], [C_12_C_1_Pip][Cl], [C_16_C_1_Pip][Cl], [C_6_C_1_Pip]_2_[LS], [C_8_C_1_Pip]_2_[LS], [C_12_C_1_Pip]_2_[LS] and [C_16_C_1_Pip]_2_[LS] were characterized by ^1^H NMR for structural confirmation. The summary of the chemical shifts is shown below. For the chloride-IL, i.e., [C_6_C_1_Pip][Cl], [C_8_C_1_Pip][Cl], [C_12_C_1_Pip][Cl] and [C_16_C_1_Pip][Cl] spectra, the chemical shifts at 0.0 to 3.00 ppm and 3.37 to 4.50 ppm represent the aliphatic chain.

[C_6_C_1_Pip][Cl], ^1^H NMR (δ, ppm in CDCl_3_): 4.12 (3H, t, N^+^C**H**_3_), 3.52 (4H, m, C**H**_2_N^+^C**H**_2_), 3.25 (4H, m, C**H**_2_CH_2_C**H**_2_), 2.71 (2H, m, N^+^C**H**_2_CH_2_), 1.73 (4H, m, CH_2_C**H**_2_CH_2_, CH_2_C**H**_2_CH_2_), 1.65 (2H, CH_2_C**H**_2_CH_2_), 1.31 (2H, m, CH_2_C**H**_2_CH_2_), 1.29 (4H, m, CH_2_C**H**_2_C**H**_2_CH_2_), 0.88 (3H, t, CH_2_C**H**_3_). Percentage yield = 87.68%.

[C_8_C_1_Pip][Cl], ^1^H NMR (δ, ppm in CDCl_3_): 4.12 (3H, t, N^+^C**H**_3_), 3.52 (4H, m, C**H**_2_N^+^C**H**_2_), 3.25 (4H, m, C**H**_2_CH_2_C**H**_2_), 2.71 (2H, m, N^+^C**H**_2_CH_2_), 1.77 (4H, m, CH_2_C**H**_2_CH_2_, CH_2_C**H**_2_CH_2_), 1.65 (2H, CH_2_C**H**_2_CH_2_), 1.31 (2H, m, CH_2_C**H**_2_CH_2_), 1.29 (8H, m, CH_2_C**H**_2_C**H**_2_C**H**_2_C**H**_2_CH2), 0.88 (3H, t, CH_2_C**H**_3_). Percentage yield = 89.91%.

[C_12_C_1_Pip][Cl], ^1^H NMR (δ, ppm in CDCl_3_): 4.01 (3H, t, N^+^C**H**_3_), 3.49 (4H, m, C**H**_2_N^+^C**H**_2_), 3.15 (4H, m, C**H**_2_CH_2_C**H**_2_), 2.86 (2H, m, N^+^C**H**_2_CH_2_), 1.75 (4H, m, CH_2_C**H**_2_CH_2_, CH_2_C**H**_2_CH_2_), 1.65 (2H, CH_2_C**H**_2_CH_2_), 1.31 (2H, m, CH_2_C**H**_2_CH_2_), 1.26 (14H, m, CH_2_C**H**_2_C**H**_2_C**H**_2_C**H**_2_C**H**_2_C**H**_2_C**H**_2_CH_2_), 0.88 (3H, t, CH_2_C**H**_3_). Percentage yield = 92.28%.

[C_16_C_1_Pip][Cl], ^1^H NMR (δ, ppm in CDCl_3_): 4.12 (3H, t, N^+^C**H**_3_), 3.52 (4H, m, C**H**_2_N^+^C**H**_2_), 3.27 (4H, m, C**H**_2_CH_2_C**H**_2_), 2.71 (2H, m, N^+^C**H**_2_CH_2_), 1.73 (4H, m, CH_2_C**H**_2_CH_2_, CH_2_C**H**_2_CH_2_), 1.70 (2H, CH_2_C**H**_2_CH_2_), 1.31 (2H, m, CH_2_C**H**_2_CH_2_), 1.21 (24H, m, CH_2_C**H**_2_C**H**_2_C**H**_2_C**H**_2_C**H**_2_C**H**_2_C**H**_2_C**H**_2_C**H**_2_C**H**_2_C**H**_2_C**H**_2_CH_2_), 0.88 (3H, t, CH_2_C**H**_3_). Percentage yield = 88.59%.

The summary of the chemical shifts for the [LS] ionic liquids [C_6_C_1_Pip]_2_[LS], [C_8_C_1_Pip]_2_[LS], [C1_2_C1Pip]_2_[LS] and [C_16_C_1_Pip]_2_[LS] is shown below and in Figure 2. The chemical shift for the aliphatic chain and hydrogen attached to the ring of each cation appeared at 0.00 to 3.24 ppm. The aliphatic chain of [LS]^2-^ had a chemical shift at 3.24 to 3.78 ppm. It is worth noting that there were 3H integrated at the chemical shift 3.78 ppm for each of the IL spectra, representing fingerprints of lignosulfonate-based ILs.

[C_6_C_1_Pip]_2_[LS], ^1^H NMR (δ, ppm in DMSO-d6): 6.11 (0H, t), 3.73 (3H, m, OC**H**3), 3.51 (1H, t, OC**H**), 3.47 (2H, d, OCHC**H**_2_S), 3.45 (2H, d, OCHC**H**_2_C), 3.43 (2H, t, CC**H**_2_C**H**_2_), 3.37 (2H, t, CH_2_C**H**_2_S), 3.33 (2H, t, CH_2_C**H**_2_CH_2_), 3.31 (8H, m, CH_2_C**H**_2_C**H**_2_CH_2_, CH_2_C**H**_2_C**H**_2_CH_2_), 3.21 (8H, t, CH_2_C**H**_2_N^+^C**H**_2_**C**H_2_, CH_2_C**H**_2_N^+^C**H**_2_**C**H_2_), 2.82 (6H, s, C**H**_3_, C**H**_3_), 2.59 (4H, t, N^+^C**H**_2_, N^+^C**H**_2_), 2.54 (4H, CH_2_C**H**_2_CH_2_, CH_2_C**H**_2_CH_2_), 2.52 (4H, m, CH_2_C**H**_2_CH_2_, CH_2_C**H**_2_CH_2_), 2.10 (4H, m, CH_2_C**H**_2_CH_2_, CH_2_C**H**_2_CH_2_), 1.53 (12H, dd, CH_2_C**H**_2_CH_2_, CH_2_C**H**_2_CH_2_), 1.12 (24H, d, CH_2_C**H**_2_CH_2_, CH_2_C**H**_2_CH_2_), 0.88 (6H, t, C**H**_3_, C**H**_3_). Percentage yield = 92.89%.

[C_8_C1Pip]_2_[LS], ^1^H NMR (δ, ppm in DMSO-d6): 6.01 (0H, t), 3.73 (3H, m, OC**H**3), 3.51 (1H, t, OC**H**), 3.47 (2H, d, OCHC**H**_2_S), 3.45 (2H, d, OCHC**H**_2_C), 3.41 (2H, t, CC**H**_2_C**H**_2_), 3.35 (2H, t, CH_2_C**H**_2_S), 3.31 (2H, t, CH_2_C**H**_2_CH_2_), 3.29 (8H, m, CH_2_C**H**_2_C**H**_2_CH_2_, CH_2_C**H**_2_C**H**_2_CH_2_), 3.23 (8H, t, CH_2_C**H**_2_N^+^C**H**_2_**C**H_2_, CH_2_C**H**_2_N^+^C**H**_2_**C**H_2_), 2.79 (6H, s, C**H**_3_, C**H**_3_), 2.53 (4H, t, N^+^C**H**_2_, N^+^C**H**_2_), 2.50 (4H, CH_2_C**H**_2_CH_2_, CH_2_C**H**_2_CH_2_), 2.49 (4H, m, CH_2_C**H**_2_CH_2_, CH_2_C**H**_2_CH_2_), 2.11 (4H, m, CH_2_C**H**_2_CH_2_, CH_2_C**H**_2_CH_2_), 1.51 (4H, dd, CH_2_C**H**_2_CH_2_, CH_2_C**H**_2_CH_2_), 1.10 (12H, d, CH_2_C**H**_2_C**H**_2_CH_2_, CH_2_C**H**_2_C**H**_2_CH_2_), 0.85 (6H, t, C**H**_3_, C**H**_3_). Percentage yield = 91.83%.

[C_12_C_1_Pip]_2_[LS], ^1^H NMR (δ, ppm in DMSO-d6): 6.31 (0H, t), 3.73 (3H, m, OC**H**3), 3.51 (1H, t, OC**H**), 3.47 (2H, d, OCHC**H**_2_S), 3.45 (2H, d, OCHC**H**_2_C), 3.43 (2H, t, CC**H**_2_C**H**_2_), 3.37 (2H, t, CH_2_C**H**_2_S), 3.33 (2H, t, CH_2_C**H**_2_CH_2_**)**, 3.31 (8H, m, CH_2_C**H**_2_C**H**_2_CH_2_, CH_2_C**H**_2_C**H**_2_CH_2_), 3.21 (8H, t, CH_2_C**H**_2_N^+^C**H**_2_**C**H_2_, CH_2_C**H**_2_N^+^C**H**_2_**C**H_2_), 2.82 (6H, s, C**H**_3_, C**H**_3_), 2.59 (4H, t, N^+^C**H**_2_, N^+^C**H**_2_), 2.54 (4H, CH_2_C**H**_2_CH_2_, CH_2_C**H**_2_CH_2_), 2.52 (4H, m, CH_2_C**H**_2_CH_2_, CH_2_C**H**_2_CH_2_), 2.10 (4H, m, CH_2_C**H**_2_CH_2_, CH_2_C**H**_2_CH_2_), 1.53 (12H, dd, CH_2_C**H**_2_C**H**_2_CH_2_, CH_2_C**H**_2_C**H**_2_CH_2_), 1.12 (24H, d, CH_2_C**H**_2_C**H**_2_C**H**_2_C**H**_2_C**H**_2_CH_2_, CH_2_C**H**_2_C**H**_2_C**H**_2_C**H**_2_C**H**_2_CH_2_), 0.88 (6H, t, C**H**_3_, C**H**_3_). Percentage yield = 89.66%.

[C_16_C_1_Pip]_2_[LS], ^1^H NMR (δ, ppm in DMSO-d6): 6.8 (0H, t), 3.78 (3H, m, OC**H**3), 3.45 (1H, t, OC**H**), 3.43 (2H, d, OCHC**H**_2_S), 3.41 (2H, d, OCHC**H**_2_C), 3.40 (2H, t, CC**H**_2_C**H**_2_), 3.39 (2H, t, CH_2_C**H**_2_S), 3.38 (2H, t, CH_2_C**H**_2_CH_2_), 3.34 (8H, m, CH_2_C**H**_2_C**H**_2_CH_2_, CH_2_C**H**_2_C**H**_2_CH_2_), 3.24 (8H, t, CH_2_C**H**_2_N^+^C**H**_2_**C**H_2_, CH_2_C**H**_2_N^+^C**H**_2_**C**H_2_), 2.99 (6H, s, C**H**_3_, C**H**_3_), 2.54 (4H, t, N^+^C**H**_2_, N^+^C**H**_2_), 2.52 (4H, CH_2_C**H**_2_CH_2_, CH_2_C**H**_2_CH_2_), 2.49 (4H, m, CH_2_C**H**_2_CH_2_, CH_2_C**H**_2_CH_2_), 2.14 (4H, m, CH_2_C**H**_2_CH_2_, CH_2_C**H**_2_CH_2_), 1.55 (16H, dd, CH_2_C**H**_2_C**H**_2_C**H**_2_C**H**_2_CH_2_, CH_2_C**H**_2_C**H**_2_C**H**_2_C**H**_2_CH_2_), 1.18 (28H, d, CH_2_C**H**_2_C**H**_2_C**H**_2_C**H**_2_C**H**_2_C**H**_2_C**H**_2_CH_2_, CH_2_C**H**_2_C**H**_2_C**H**_2_C**H**_2_C**H**_2_C**H**_2_C**H**_2_CH_2_), 0.85 (6H, t, C**H**_3_, C**H**_3_). Percentage yield = 93.46%.

### 2.3. Thermal Gravimetric Analysis (TGA)

The thermal stability of synthesized lignosulfonate-based ILs was analyzed by TGA as shown in Figure 3 and Table 4. The onset degradation of [C_16_C_1_Pip]_2_[LS], [C_12_C_1_Pip]_2_[LS], [C_8_C1Pip]_2_[LS] and [C_6_C_1_Pip]_2_[LS] was at temperatures of 208.83 °C, 204.87 °C, 160.70 °C and 123.24 °C, respectively.

The first stage degradation of [C_16_C_1_Pip]_2_[LS], [C_12_C_1_Pip]_2_[LS], [C_8_C1Pip]_2_[LS] and [C_6_C_1_Pip]_2_[LS] was at temperature ranges of, respectively, 208.83–273.19 °C, 204.87–254.67 °C, 160.70–235.04 °C and 118.24–207.21 °C. Labile functional groups such as sulfonyl, carboxylate [56], methoxy [57], methyl, hydroxyl and carbonyl, which are connected in straight-link [51,52], are likely decomposed in this stage.

At temperature ranges of 273.19–453.78 °C, 254.67–375.20 °C, 235.04–333.30 °C and 207.21–305.49 °C, the second stage of degradation occurred, where side aliphatic chains and aliphatic rings of cations [58] for the respective [C_16_C_1_Pip]_2_[LS], [C_12_C_1_Pip]_2_[LS], [C_8_C1Pip]_2_[LS] and [C_6_C_1_Pip]_2_[LS] were degraded. The longer side alkyl chain of piperidinium is attributed to a higher degradation temperature. The side alkyl chain of cations contributed to the thermal stability of ILs.

The third stage degradation of [C_16_C_1_Pip]_2_[LS], [C_12_C_1_Pip]_2_[LS], [C_8_C1Pip]_2_[LS] and [C_6_C_1_Pip]_2_[LS] was at temperature ranges of 453.78–800 °C, 375.20–800 °C and 333.33–305.49 °C. In this stage, the backbone of ILs incorporated with polynuclear aromatic rings was degraded [59]. However, one can see in Figure 3, at a temperature of 800 °C, that there were 43.14%, 32.62%, 28.74% and 23.09% of ILs remaining for the respective [C_16_C_1_Pip]_2_[LS], [C_12_C_1_Pip]_2_[LS], [C_8_C1Pip]_2_[LS] and [C_6_C_1_Pip]_2_[LS]. Beyond a temperature of 790 °C, a biomass-based material such as lignosulfonate is converted into carbon soot, which possesses a high capacity to be decomposed [60]. Domínguez et al. reported that the sample presents high thermal stability when the value of carbon soot content is high [61].

### 2.4. Asphaltene Dispersion Study

An asphaltene stock solution was prepared and a standard calibration curve was obtained. Asphaltene dispersion studies involved the onset precipitation studies, contact time, temperature effect and the effect of IL concentration.

#### 2.4.1. Asphaltene Onset Precipitation

The synthesized ILs were tested for asphaltene onset precipitation. [C_6_C_1_Pip][Cl], [C_8_C_1_Pip][Cl], [C_12_C_1_Pip][Cl], [C_16_C_1_Pip][Cl], [C_6_C_1_Pip]_2_[LS], [C_8_C_1_Pip]_2_[LS], [C_12_C1Pip]_2_[LS] and [C_16_C_1_Pip]_2_[LS] were added into different volume ratios of *n*-heptane and 100 ppm asphaltene solution. The onset of asphaltene aggregation was reported to occur at a concentration of approximately ≈100 ppm [36]. Figure 4 shows that the ILs delayed the asphaltene precipitation at a 1:1 *n*-heptane–toluene ratio. At 1:1 *n*-heptane–toluene ratio, [C_6_C_1_Pip][Cl] depicted the highest absorbance at 346 nm wavelength compared to [C_16_C_1_Pip]_2_[LS]. The order of the ILs’ performance on the asphaltene onset precipitation test is as follows: [C_16_C_1_Pip]_2_[LS] > [C_12_C_1_Pip]_2_[LS] > [C_8_C_1_Pip]_2_[LS] > [C_6_C_1_Pip]_2_[LS] > [C_16_C_1_Pip][Cl] > [C_12_C_1_Pip][Cl] > [C_8_C_1_Pip][Cl] > [C_6_C_1_Pip][Cl]. The combination of the lignosulfonate anion can give a better delay precipitation of asphaltene. [C_16_C_1_Pip]_2_[LS] can delay the precipitation of asphaltene better than [C_6_C_1_Pip][Cl]. The longer alkyl chain attached in the cation head of [C_16_C_1_Pip]_2_[LS] and ILs with the combination of the lignosulfonate anion were attributed to a better steric hindrance effect, helping to inhibit the self-association of asphaltene molecules [12,62].

#### 2.4.2. Contact Time Effect

This study aimed to investigate the performance of synthesized ILs in asphaltene dispersion. The IL was mixed with the model oil and the contact time varied, i.e., 0, 10, 20, 30, 40, 50, 60, 70 and 80 min. As shown in Figure 5, a higher asphaltene dispersion index can be achieved when applying a longer contact time between ILs and the model oil. A longer contact time provided a high capacity for lignosulfonate-based ILs to form intermolecular forces via electrostatic interaction and π–π interaction with asphaltene molecules [29]. The following order of dispersion index for contact time effect was observed: [C_16_C_1_Pip]_2_[LS] > [C_12_C1Pip]_2_[LS] > [C_8_C1Pip]_2_[LS] > [C_6_C1Pip]_2_[LS] > [C_16_C_1_Pip][Cl] > [C_12_C_1_Pip][Cl] > [C_8_C_1_Pip][Cl] > [C_6_C_1_Pip][Cl]. [C_6_C_1_Pip][Cl] depicted the lowest dispersion index, while [C_16_C_1_Pip]_2_[LS] exhibited the highest dispersion index at a contact time of 80 min. The alkyl chain with the highest number of carbons and the presence of a lignosulfonate anion can, respectively, increase the hydrophobicity and hydrophilicity of [C_16_C_1_Pip]_2_[LS], thus disrupting intermolecular force interactions between asphaltene molecules.

#### 2.4.3. Temperature Effect

This study was carried out by varying the model oil temperature. The synthesized lignosulfonate-based ILs were mixed with model oils. As shown in Figure 6, the ILs possessed a high dispersion index despite the high temperature. The following order of asphaltene dispersion index performance for the temperature test was observed: [C_16_C_1_Pip]_2_[LS] > [C_12_C_1_Pip]_2_[LS] > [C_8_C1Pip]_2_[LS] > [C_6_C_1_Pip]_2_[LS] > [C_16_C_1_Pip][Cl] > [C_12_C_1_Pip][Cl] > [C_8_C_1_Pip][Cl] > [C_6_C_1_Pip][Cl]. This is because the presence of polynuclear aromatic groups in the ILs contributes to the thermal stability of ILs, eventually fostering the π–π interaction with asphaltene molecules to improve the stabilization of asphaltene [12,37]. The [C_6_C_1_Pip][Cl] dispersion index at various temperatures is the lowest, as the IL incorporated with the shortest alkyl chain is the least stable when the temperature effect is applied. A lignosulfonate-based IL with a high number of carbons, [C_16_C_1_Pip]_2_[LS], can achieve a high dispersion index at every temperature applied during the test. [C_16_C_1_Pip]_2_[LS] possessed high thermal stability according to a thermal gravimetric analysis study. The high thermal stability of [C_16_C_1_Pip]_2_[LS] ensured that vital functional groups contributing to the interaction towards asphaltene molecules were not being decomposed. Hence, the higher number of carbons in ILs not only increases the non-polarity of the ILs towards asphaltene molecules and non-polar media but is attributed to conducive thermal stability towards IL.

#### 2.4.4. IL Concentration Effect

This study was carried out at a temperature of 95 °C in a reflux condenser and the concentrations of respective ILs varied, being 10,000, 20,000, 30,000, 40,000 and 50,000 ppm. The results are shown in Figure 7. The dispersion index of the ILs was higher when the concentration of the ILs was higher as well. A higher concentration of ILs can provide a higher interaction capacity and non-polarity medium towards asphaltene molecules. The following order of asphaltene dispersion index performance for the IL concentration effect was observed: [C_16_C_1_Pip]_2_[LS] > [C_12_C_1_Pip]_2_[LS] > [C_8_C1Pip]_2_[LS] > [C_6_C_1_Pip]_2_[LS] > [C_16_C_1_Pip][Cl] > [C_12_C_1_Pip][Cl] > [C_8_C_1_Pip][Cl] > [C_6_C_1_Pip][Cl]. The dispersion indexes of [C_16_C_1_Pip][Cl], [C_12_C_1_Pip][Cl], [C_8_C_1_Pip][Cl], [C_6_C_1_Pip][Cl], [C_12_C_1_Pip]_2_[LS], [C_8_C1Pip]_2_[LS] and [C_6_C_1_Pip]_2_[LS] were significantly lower compared to [C_16_C_1_Pip]_2_[LS]. This implied that a longer piperidinium alkyl chain length and the presence of the lignosulfonate anion affected the dispersity of asphaltene molecules. The increase in alkyl chain length promoted a van der Waal’s force towards asphaltene molecules since the hydrophobicity and non-polarity of the ILs increased. The high number of carbons in the IL depicted a large non-polar surface which caused less compactness of the IL itself, eventually providing the optimum environment for asphaltene dissolution.

### 2.5. Asphaltene Aggregation Study

Referring to the asphaltene dispersion study, [C_16_C_1_Pip]_2_[LS] possessed the highest dispersion index for respective contact time effect, temperature effect and ILs concentration effect studies. It implied that [C_16_C_1_Pip]_2_[LS] can disperse asphaltene molecules and inhibit asphaltene aggregation growth in model oil more effectively compared to other synthesized ILs. So, [C_16_C_1_Pip]_2_[LS] inhibition performance was tested in the asphaltene aggregation study. The asphaltene aggregation study involved asphaltene particle size distribution and particle size diameter studies.

#### 2.5.1. Asphaltene Particle Size Distribution

The particle size distribution of asphaltene was analyzed using the dynamic light scattering method. The model oil was treated with [C_16_C_1_Pip]_2_[LS]. The results of the particle size distribution are shown in Figure 8. The particle size of asphaltenes reduced with the addition of [C_16_C_1_Pip]_2_[LS] as the aggregate’s polydispersity was high. The intensity of the >200 nm particle size is lower than the <200 nm particle size. It can be concluded that [C_16_C_1_Pip]_2_[LS] can inhibit aggregate growth into a lower than 200nm particle size. [C_16_C_1_Pip]_2_[LS] exhibited an asphaltene dispersant effect by forming a network of interactions including Coulombic interactions, π–π interactions, hydrogen bonds and dipole–dipole interactions.

#### 2.5.2. Asphaltene Particle Size Diameter

The particle size diameter of asphaltenes in model oil was analyzed using the dynamic light scattering method after the model oil was treated with [C_16_C_1_Pip]_2_[LS]. Different concentrations of [C_16_C_1_Pip]_2_[LS] were used during the particle size diameter test. Figure 9 indicates the impact of different [C_16_C_1_Pip]_2_[LS] concentrations on aggregate diameter. It can be observed that the IL can significantly inhibit aggregate growth until a concentration of 40,000 ppm is reached. The [C_16_C_1_Pip]_2_[LS] possessed an asphaltene inhibitor property as the lignosulfonate-based IL was able to interact with asphaltene molecules and restrict precipitation.

### 2.6. Kinetic Study

Referring to asphaltene dispersion performance and asphaltene aggregation studies, [C_16_C_1_Pip]_2_[LS], respectively, possessed high dispersion index performance as compared to other synthesized ILs, reduced intensity of >200 nm asphaltene particle size and inhibited asphaltene particle growth at the maximum concentration of IL. So, [C_16_C_1_Pip]_2_[LS] was tested in kinetic study models to determine a lead mechanism interaction involved during asphaltene dispersion.

In this study, the amount of adsorbed asphaltene at 100 pm as a function of contact time was studied. The experiment was carried out at 24.7 °C using 10,000 mg/L IL. Figure 10 indicates that the removal efficiency significantly increased from zero min to sixty min of contact time. After 60 min, the removal efficiency halted, from 63.20% to 62.74%, at a contact time of 100 min.

Figure 11 shows that the asphaltene removal capacity increased from zero min to sixty min of contact time, and slightly decreased after sixty min. The results implied that the completed adsorption occurred within 60 min. The equilibrium efficiency and capacity of [C_16_C_1_Pip]_2_[LS] to remove asphaltene were achieved within 60 min. The figures reveal that the asphaltene equilibrium uptake on [C_16_C_1_Pip]_2_[LS] was 6.41 mg/g at a contact time of 60 min. This implied that there were high molecular interactions [63] between asphaltene aggregates and IL.

Table 5 and Figure 12 summarizes the obtained kinetic parameters for linear pseudo-first-order and linear pseudo-second-order kinetic models. Based on the results, the pseudo-second-order is more favorable for the experimental data of asphaltene adsorption on [C_16_C_1_Pip]_2_[LS] as the R^2^ = 0.9577 was higher than the R^2^ = 0.8652 of the linear pseudo-first order model. This implies that chemical adsorption is the leading mechanism of asphaltene adsorption by [C_16_C_1_Pip]_2_[LS].

### 2.7. HOMO–LUMO Study

Asphaltene dispersion and asphaltene aggregation studies implied that [C_16_C_1_Pip]_2_[LS] was a good asphaltene dispersant and asphaltene growth inhibitor. Chemical adsorption was the lead mechanism of asphaltene adsorption. Thus, the HOMO–LUMO study was carried out to explore the chemical interaction mechanism as well as the interactive nature of asphaltene towards the IL, [C_16_C_1_Pip]_2_[LS]. The interaction between molecules happened between the HOMO of a molecule and the LUMO of a molecule. The amount of energy required to add electrons in a molecule can be obtained from the HOMO. The amount of energy required to remove electrons in a molecule can be obtained from LUMO [64].

HOMO–LUMO energy gap can be obtained from the respective HOMO and LUMO energy values. A high HOMO–LUMO energy gap is attributed to high resistance to charge transfer and hard molecules. A low HOMO–LUMO energy gap is attributed to soft molecules and high polarizability [65].

With [C_16_C_1_Pip]_2_[LS] in Figure 13, HOMO energies originate from the heteroatoms of the anion and LUMO energies originate from the heteroatoms of the cations and the centers of aromatic rings. The HOMO bulk of energies originates from O=S(-O-)=O, while the LUMO bulk of energies originates from CH_2_-N^+^-CH_2_ and C=C groups. The anions can act as electron donors, and cations have a high electron affinity towards asphaltene [66]. The anions of [C_16_C_1_Pip]_2_[LS] characterized the nucleophilicity, while its cations characterized the electrophilicity of an IL.

One can see that the HOMO and LUMO states of the asphaltene model in Figure 14 were mainly localized on the central aromatic rings. The HOMO energies originated from the heteroatoms of nitrogen and sulfur. The LUMO energies originated from aromatic rings. Nitrogen and sulfur exhibited the possibility to form a hydrogen bond, while aromatic rings can form π–π interactions with other molecules. The nitrogen and sulfur parts of the asphaltene model characterized the nucleophilicity, while the aromatic rings characterized the electrophilicity of the model.

As shown in Figure 15, HOMO bulky energies originated from the anionic center of IL, while the LUMO adducts originated from the asphaltene model. We can conclude that IL acted as a nucleophile, and the asphaltene model possessed electrophilicity. The electron donation interaction from IL towards the asphaltene model can occur. Since the interaction between IL and the asphaltene model was HOMO to LUMO, this resulted in adduct formation controlled by hydrogen bonding towards the heteroatoms of the asphaltene model.

According to Table 6, the HOMO–LUMO gap energy of [C_16_C_1_Pip]_2_[LS] was 0.255 eV. [C_16_C_1_Pip]_2_[LS] possessed a small HOMO–LUMO gap energy, concluding it is a soft molecule and has better polarizability, as observed in dispersion studies. The HOMO–LUMO gap energy of asphaltene was 1.279 eV. A high HOMO–LUMO gap energy exhibits high resistance to changes in electron number and distribution. Thus, asphaltene was the hardest model molecule as compared to IL. Adducts formed when [C_16_C_1_Pip]_2_[LS] was employed, indicative that the HOMO comes from IL’s anionic center and the LUMO originates from the asphaltene model. It implied the presence of electron donation from the HUMO to the LUMO. Hence, there is a possible interaction resulting in adduct formation controlled by hydrogen bonding, as well as the π face of the asphaltene. The HOMO–LUMO gap energy of [C_16_C_1_Pip]_2_[LS] + asphaltene was 0.089 eV. The orbital energy gaps of individual IL and asphaltene were higher as compared to IL + asphaltene. The HOMO–LUMO gaps were observed in the following order: asphaltene > [C_16_C_1_Pip]_2_[LS] > [C_16_C_1_Pip]_2_[LS] + asphaltene. This implied that [C_16_C_1_Pip]_2_[LS] + asphaltene were easier to polarize, had higher solubility, and higher asphaltene dispersion. In addition, [C_16_C_1_Pip]_2_[LS] became lower in LUMO energy, −2.111 eV compared to the individual LUMO of IL, resulting from adducts formed between asphaltene and the IL (Table 6). Therefore, there was possible overlap of the LUMO sites with incoming HOMO orbitals (Figure 15).

## 3. Materials and Methods

### 3.1. Materials

Lignosulfonic acid sodium salt [Na]_2_[LS] (99.99%), *n*-methyl-piperidine (99.00%), 1-chlorohexane (99.00%), 1-chlorooctane, 1-chlorododecane, 1-chlorohexadecane (97.00%), *n*-heptane, toluene and acetonitrile (≥99.50%) were purchased from Merck and Sigma and were used as received.

### 3.2. Methods

#### 3.2.1. Quaternization of *N*-Methyl-Piperidine

A 250 mL three-necked flask was equipped with a condenser and stirrer. A total of 0.05 mol *N*-methyl-piperidine was dissolved in 30 mL of acetonitrile. Then, 0.06 mol of 1-chlorohexane was added to the reaction mixture and further stirred at 400 rpm, at a temperature of 80 °C for 72 h. The solvent was removed using a rotary evaporator under reduced temperature until a brown viscous liquid appeared after cooling to room temperature. The liquid was dissolved in chloroform; water was added and extracted with chloroform 2 times. The solvent in combined organic layers was removed using a rotary evaporator under reduced pressure and dried for 24 h to give the [C_6_C_1_Pip][Cl]. The synthesis was repeated with different haloalkanes with different side alkyl chains: 1-chlorooctane, 1-chlorododecane and 1-chlorohexadecane (Figure 16).

#### 3.2.2. Metathesis of Lignosulfonate-Based Ionic Liquids

The synthesis of [RC_1_Pip]_2_[LS] was carried out using a 500 mL 3-neck round-bottom flask, heating plate and reflux condenser. Then, 0.05 mmol of [Na]_2_[LS] and 0.10 mmol of [RC_1_Pip][Cl] (R = C_6_, C_8_, C_12_, C_16_) were mixed in 100 mL intermediate solvent of water, and were further stirred for 400 rpm at a temperature of 55 °C for 72 h. The solvent was removed using a rotary evaporator until a dark brown viscous solid appeared. The brown viscous solid was placed in the freezer, and then a colorless crystal appeared indicating the presence of a by-product of sodium chloride, [Na][Cl]. Acetonitrile was used to solidify the sodium chloride [67] until the crystal disappeared when the solid was placed in the freezer. The brown solid was further dried using a rotary evaporator for 3 h and oven for 24 h to form a viscous solid of [RC_1_Pip]_2_[LS] (Figure 17).

#### 3.2.3. Extraction of Asphaltene from Crude Oil

A total of 200 mL of *n*-heptane and 5 g of crude oil were measured and weighed, respectively, in a 250 mL round bottom flask. The resulting mixture was sonicated for 3 h and then left for 24 h. The insoluble fraction was then filtered using 70 mm Advantec glass fiber and collected. The fraction was placed in a Soxhlet with *n*-heptane for 48 h. The precipitate was removed, weighed and placed in the Soxhlet until a constant weight was obtained. The precipitate was then dried in the vacuum oven until a constant weight was achieved.

#### 3.2.4. Fourier-Transform Infrared Attenuated Total Reflection (FTIR-ATR)

In total, 20 mg samples of [RC_1_Pip][Cl], [Na]_2_[LS] and [RC_1_Pip]_2_[LS] were placed in sample vials and dried in the vacuum oven to reduce moisture for 24 h. The FTIR spectra of the samples were recorded using PerkinElmer Spectrum in the range of 4000 cm^−1^ to 500 cm^−1^ at 4 cm^−1^ resolution by PerkinElmer Spectrum Frontier.

#### 3.2.5. ^1^H NMR

A total of 100 µL of [RC_1_Pip][Cl] and [RC_1_Pip]_2_[LS] were mixed in deuterated chloroform (CDCl_3_) and deuterated dimethyl sulfoxide (C_2_D_6_OS), respectively, in the sample vials. The vials were swirled gently. Each of the samples was carefully transferred into the NMR tube. The height of each sample was ensured to be approximately the same as the reference NMR tube. The samples were placed on a sample deck. ^1^H NMR spectra of the samples were recorded and processed using Bruker TopSpin 3.2 by Bruker AscendTM500 (Bruker UK Limited, Coventry, UK).

#### 3.2.6. TGA

A total of 25 mg of dried [RC_1_Pip]_2_[LS] sample was placed on the sample deck. The sample was analyzed between the temperature range of 24 °C to 700 °C at a heating rate of 10 °C/min in Perkin Elmer STA6000.

#### 3.2.7. Preparation of Asphaltene Standard Solution Using UV-Vis

In total, 100 mg of dried asphaltene solid was weighed and diluted by 100 mL solvent of toluene to prepare the 1000 ppm asphaltene stock solution. The stock solution was diluted into 100 ppm, 90 ppm, 80 ppm, 70 ppm, 60 ppm, 50 ppm, 40 ppm, 30 ppm, 20 ppm and 10 ppm. Standard calibration was obtained using UV-Vis data at a wavelength of 346 nm using Agilent Cary100.

#### 3.2.8. Asphaltene Onset Precipitation of [RC_1_Pip]_2_[LS] and [RC_1_Pip][Cl]

There is a linear relationship between the absorbance of light and the amount of asphaltene in the toluene solution. The calibration curve was established at a wavelength of 346 nm. Then, 100 ppm of asphaltene solution was used to prepare the model oil. The solution was diluted by *n*-heptane solvent with different volume ratios (*v*/*v*). A total of 0.1 g of synthesized [RC_1_Pip]_2_[LS] and [RC_1_Pip][Cl] were, respectively, mixed in 10 mL of model oil. Then, 10,000 ppm of synthesized [RC_1_Pip]_2_[LS] and [RC_1_Pip][Cl] were, respectively, placed into each of the model oils. The samples were centrifuged at 3000 rpm for 15 min. The samples were filtered using a 0.45 µm syringe filter to remove any traces of dust. Asphaltene onset precipitation between the additive-free model oil and IL-additive model oil was compared.

#### 3.2.9. Contact Time Effect of [RC_1_Pip]_2_[LS] and [RC_1_Pip][Cl] on Asphaltene Model Oil

The volume ratio of *n*-heptane/100 ppm asphaltene solution 2:1 (*v*/*v*) was used for the contact time test. In total, 0.1 g of synthesized [RC_1_Pip]_2_[LS] and [RC_1_Pip][Cl] were, respectively, mixed in 10 mL of the model oil. Then, 10,000 ppm of synthesized [RC_1_Pip]_2_[LS] and [RC_1_Pip][Cl] were, respectively, prepared after being mixed into each of the model oils under ambient temperature. The time taken for this varied, being 0, 20, 40, 60 or 80 min. The samples were centrifuged at 4000 rpm for 15 min and filtered using a 0.45 µm syringe filter.

#### 3.2.10. Temperature Effect of [RC_1_Pip]_2_[LS] and [RC_1_Pip][Cl] on Asphaltene Model Oil

The same ratio of asphaltene model oil was used. In total, 0.1 g of synthesized ILs were, respectively, mixed in 10 mL of model oil. Then, 10,000 ppm of ILs were, respectively, prepared in each of the model oils. Each mixture was stirred for 80 min. The temperature of the mixtures varied, being 24, 40, 60, 80 or 100 °C. The samples were centrifuged at 4000 rpm for 15 min and filtered using a 0.45 µm syringe filter.

#### 3.2.11. [RC_1_Pip]_2_[LS] and [RC_1_Pip][Cl] Concentration Effect on Asphaltene Model Oil

The same model oil was used for this test. Overall, 0.1 g, 0.2 g, 0.3 g, 0.4 g and 0.5 g of the synthesized ILs were, respectively, mixed in 10 mL of model oil. Then, different concentrations of ILs were, respectively, prepared. These concentrations varied, being 10,000, 20,000, 30,000, 40,000, or 50,000 ppm in model oil. Contact time and temperature were constant at 80 min and 100 °C, respectively. The samples were then centrifuged at 4000 rpm for 15 min and filtered using a 0.45 µm syringe filter.

#### 3.2.12. Dispersion Calculation

The absorbance of model oil after being treated with IL was measured with a UV spectrophotometer at a wavelength of 346 nm. The percentage of the dispersion index is defined by this equation [20,40]:(1)$$Dispersion\;index=\frac{A-Ae}{A}\times 100\%$$
where A represents the absorbance of asphaltenes in toluene without the addition of any dispersant and Ae represents the absorbance of asphaltenes precipitate left after the dispersion. This criterion provides an estimate of the asphaltenes that would have been precipitated or suspended in the solution.

#### 3.2.13. Particle Size Analysis

The sizes and size distributions of the asphaltene’s dispersion were measured by the dynamic light scattering methods using a Malvern Zetasizer Nano ZS light scattering instrument (Malvern Instruments Ltd., Malvern, Worcester, UK) with a 532 nm laser. Measurements were conducted under a temperature condition of 25 ± 0.2 °C. The dispersed asphaltenes were evaluated with the aid of Malvern DTS software.

#### 3.2.14. Kinetic Study

Linear pseudo-first-order and pseudo-second-order were used to determine the possessed interaction mechanism. The equation of the kinetic model, respectively, was given by Equations (2) and (3) [38,42]:(2)$$\ln\left(Q_{eq}-Q_{t}\right)=\ln\left(Q_{eq}\right)-K_{t}$$
(3)$$\frac{t}{Q_{t}}=\frac{1}{KQ_{eq}^{2}}+\frac{1}{Qeq}$$ K is the equation constant, Q_eq_ is the amount of asphaltene adsorbed at equilibrium, Q_t_ is the amount of asphaltene adsorbed at respective time and t is the time taken for asphaltene adsorption.

#### 3.2.15. HOMO–LUMO Study

The structures with the optimized chemical structure and quantum calculation for molecular architectures were made using Turbomole-X (TMoleX) software package, Version 4.1.1. Quantum calculations were carried out on the density functional theory level (DFT) with a resolution identity (RI) approximation. Triple zeta valence polarized (TZVP) basis set and Becke, 3-parameter, Lee-Yang-Parr (B3-LYP) basis set were used for energy calculations. HOMO, LUMO and HOMO–LUMO orbital energy gap were calculated.

## 4. Conclusions

1-Hexyl-1-methyl-piperidinium lignosulfonate [C_6_C_1_Pip]_2_[LS], 1-octyl-1-methyl-piperidinium lignosulfonate [C_8_C_1_Pip]_2_[LS], 1-dodecyl-1-methyl-piperidinium lignosulfonate [C_12_C_1_Pip]_2_[LS] and 1-hexadecyl-1-methyl-piperidinium lignosulfonate [C_16_C_1_Pip]_2_[LS] were successfully synthesized and characterized. The presence of the vital band assignment C-O stretch of methoxy in FTIR and 3H at a chemical shift of 3.78 in ^1^H NMR as fingerprints of lignosulfonate-based material depicted successful synthesis. [RC_1_Pip]_2_[LS] has an onset degradation temperature range of 123.24–208.83 °C and the offset temperature is 800 °C. 1-Hexadecyl-1-methyl-piperidinium lignosulfonate [C_16_C_1_Pip]_2_[LS] achieved a maximum asphaltene dispersion index of 91.2% at a maximum concentration of 50,000 ppm. At the maximum concentration of 50,000 ppm of [C_16_C_1_Pip]_2_[LS], the IL successfully inhibits asphaltene aggregate growth at 11 nm. The leading interaction mechanism between [C_16_C_1_Pip]_2_[LS] and asphaltene molecules was chemical adsorption because the R^2^ = 0.9577, favoring the linear pseudo-second-order kinetic model. The HOMO–LUMO orbital energy gap study was in agreement with the combined dispersion study, aggregation study, and kinetic study model. A small HOMO–LUMO orbital energy gap can be achieved when [C_16_C_1_Pip]_2_[LS] + asphaltene model are combined, showing that the interaction is highly polarizable. There was possible electron donation interaction from the HOMO to the LUMO site. The interaction was possibly a result of hydrogen bonding and π–π interaction.

## Figures and Tables

**Figure 1 molecules-28-03390-f001:**
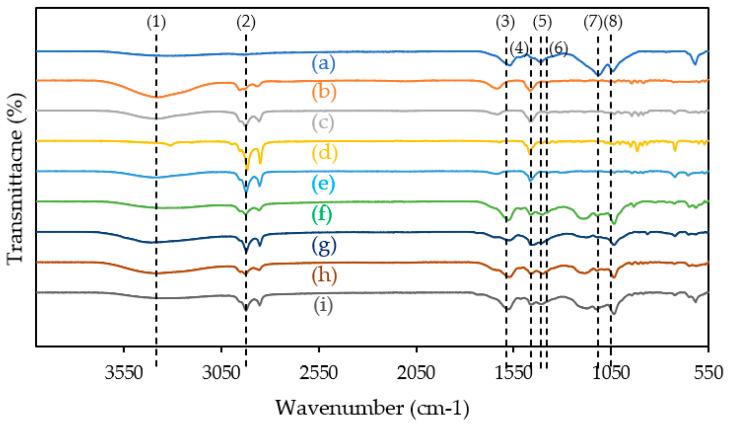
FTIR spectrum of (a) [Na]_2_[LS], (b) [C_6_C_1_Pip][Cl], (c) [C_8_C_1_Pip][Cl], (d) [C_12_C_1_Pip][Cl], (e) [C_16_C_1_Pip][Cl], (f) [C_6_C_1_Pip]_2_[LS], (g) [C_8_C_1_Pip]_2_[LS], (h) [C_12_C_1_Pip]_2_[LS] and (i) [C_16_C_1_Pip]_2_[LS]. Wavenumber peaks (1) 3350 cm^−1^, (2) 2930 cm^−1^, (3) 1560 cm^−1^, (4) 1460 cm^−1^, (5) 1400 cm^−1^, (6) 1350 cm^−1^, (7) 1110 cm^−1^ and (8) 1050 cm^−1^.

**Figure 2 molecules-28-03390-f002:**
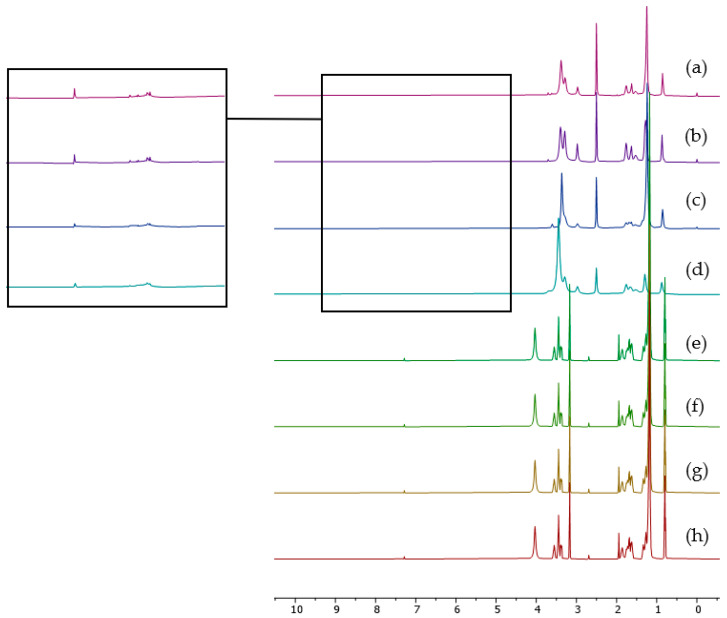
NMR spectra of synthesized cations and ILs. (a) [C_6_C_1_Pip]_2_[LS], (b) [C_8_C1Pip]_2_[LS], (c) [C_12_C_1_Pip]_2_[LS], (d) [C_16_C_1_Pip]_2_[LS], (e) [C_6_C_1_Pip][Cl], (f) [C_8_C_1_Pip][Cl], (g) [C_12_C_1_Pip][Cl], (h) [C_16_C_1_Pip][Cl].

**Figure 3 molecules-28-03390-f003:**
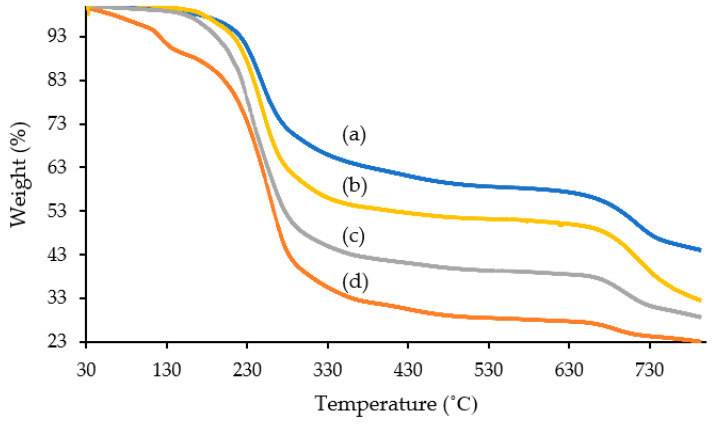
Thermal gravimetric analysis study of synthesized lignosulfonate-based ILs. (a) [C_16_C_1_Pip]_2_[LS], (b) [C_12_C_1_Pip]_2_[LS], (c) [C_8_C1Pip]_2_[LS], (d) [C_6_C_1_Pip]_2_[LS].

**Figure 4 molecules-28-03390-f004:**
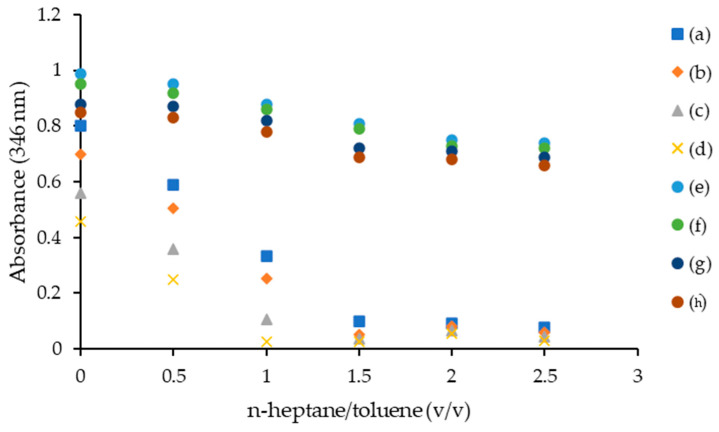
Asphaltene onset precipitation. (a) [C_6_C_1_Pip]_2_[LS], (b) [C_8_C_1_Pip]_2_[LS], (c) [C_12_C1Pip]_2_[LS], (d) [C_16_C_1_Pip]_2_[LS], (e) [C_6_C_1_Pip][Cl], (f) [C_8_C_1_Pip][Cl], (g) [C_12_C_1_Pip][Cl], (h) [C_16_C_1_Pip][Cl].

**Figure 5 molecules-28-03390-f005:**
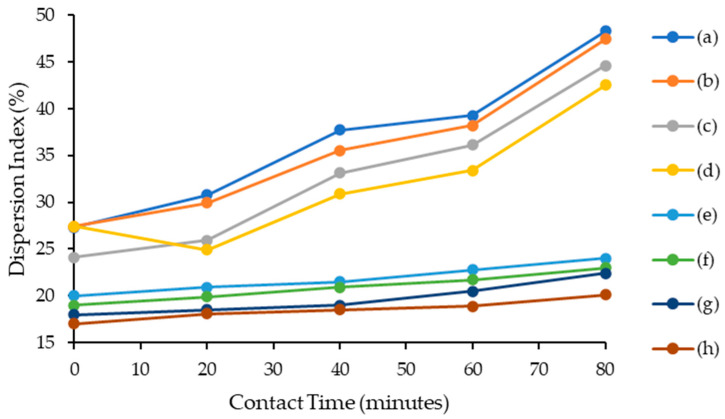
Contact time effect for asphaltene dispersion study. (a) [C_16_C_1_Pip]_2_[LS], (b) [C_12_C_1_Pip]_2_[LS], (c) [C_8_C1Pip]_2_[LS], (d) [C_6_C_1_Pip]_2_[LS] (e) [C_16_C_1_Pip][Cl], (f) [C_12_C_1_Pip][Cl], (g) [C_8_C_1_Pip][Cl], (h) [C_6_C_1_Pip][Cl].

**Figure 6 molecules-28-03390-f006:**
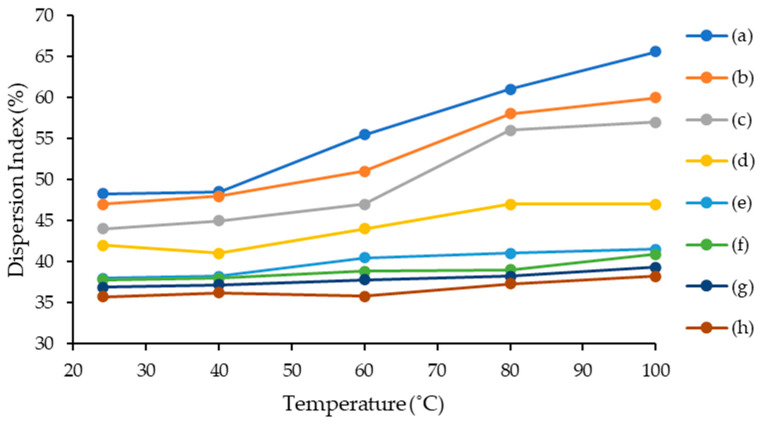
Temperature effect for asphaltene dispersion study. (a) [C_16_C_1_Pip]_2_[LS], (b) [C_12_C_1_Pip]_2_[LS], (c) [C_8_C1Pip]_2_[LS], (d) [C_6_C_1_Pip]_2_[LS] (e) [C_16_C_1_Pip][Cl], (f) [C_12_C_1_Pip][Cl], (g) [C_8_C_1_Pip][Cl], (h) [C_6_C_1_Pip][Cl].

**Figure 7 molecules-28-03390-f007:**
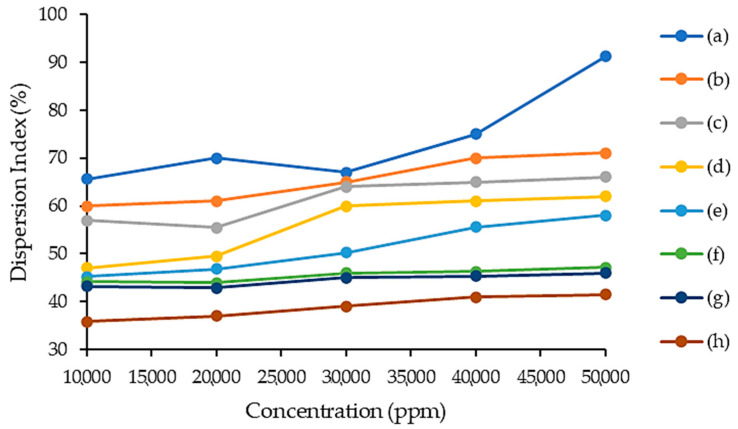
IL concentration effect for asphaltene study. (a) [C_16_C_1_Pip]_2_[LS], (b) [C_12_C_1_Pip]_2_[LS], (c) [C_8_C1Pip]_2_[LS], (d) [C_6_C_1_Pip]_2_[LS] (e) [C_16_C_1_Pip][Cl], (f) [C_12_C_1_Pip][Cl], (g) [C_8_C_1_Pip][Cl], (h) [C_6_C_1_Pip][Cl].

**Figure 8 molecules-28-03390-f008:**
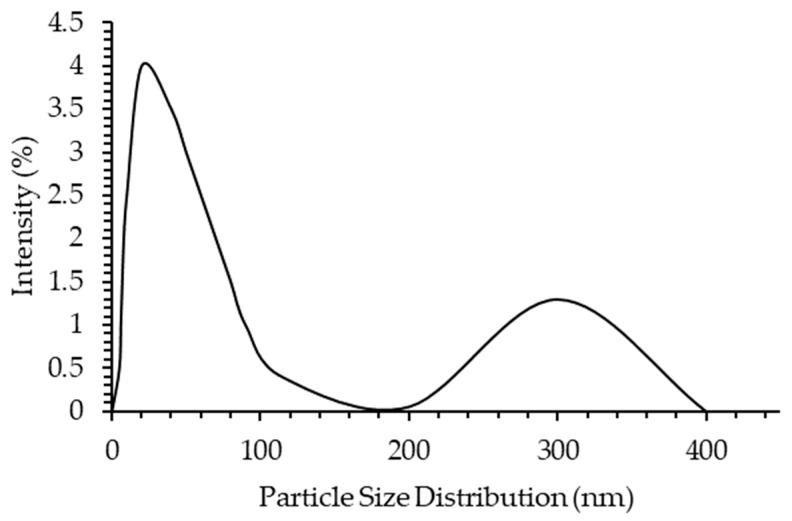
Particle size distribution. [C_16_C_1_Pip]_2_[LS] treated model oil.

**Figure 9 molecules-28-03390-f009:**
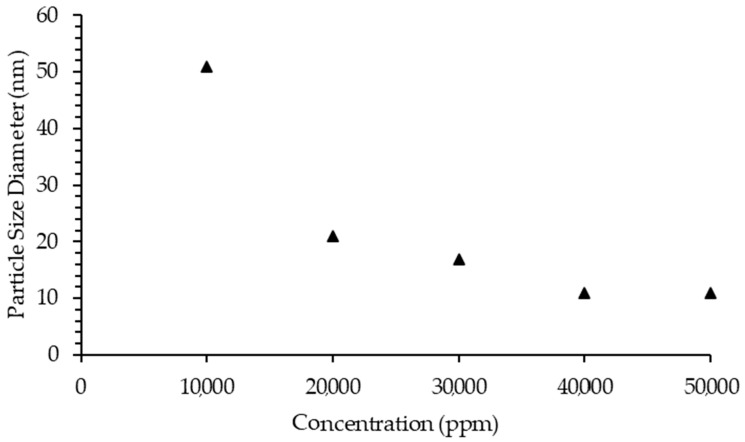
Particle size diameter [C_16_C_1_Pip]_2_[LS] treated model oil. The triangle refer to “asphaltene particle size diameter”.

**Figure 10 molecules-28-03390-f010:**
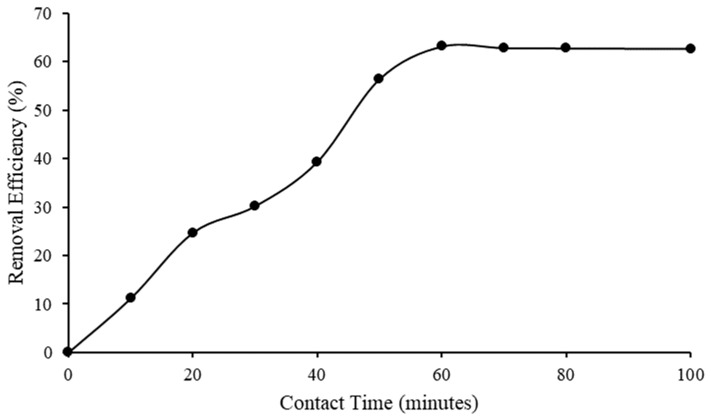
Asphaltene removal efficiency after being treated with [C_16_C_1_Pip]_2_[LS].

**Figure 11 molecules-28-03390-f011:**
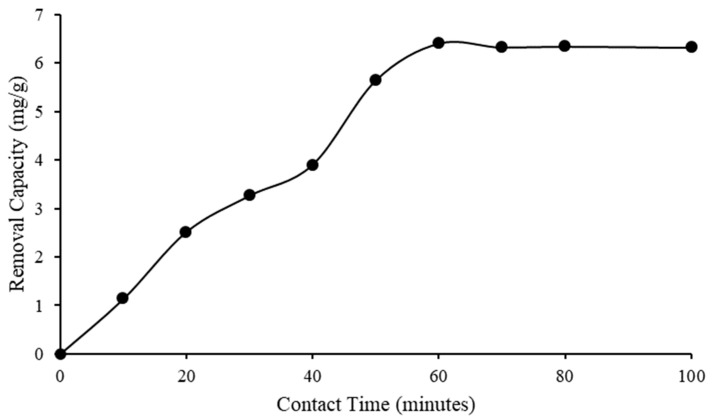
Asphaltene removal capacity after being treated with [C_16_C_1_Pip]_2_[LS].

**Figure 12 molecules-28-03390-f012:**
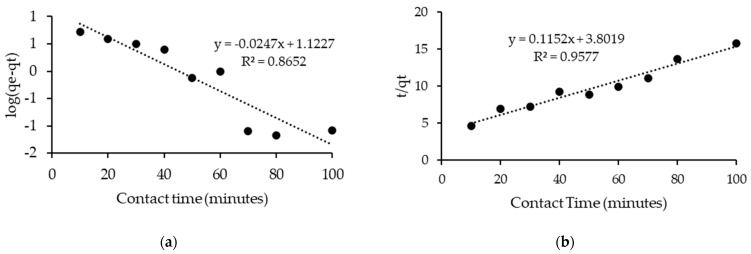
(**a**) Linear pseudo-first order, (**b**) Linear pseudo-second order.

**Figure 13 molecules-28-03390-f013:**
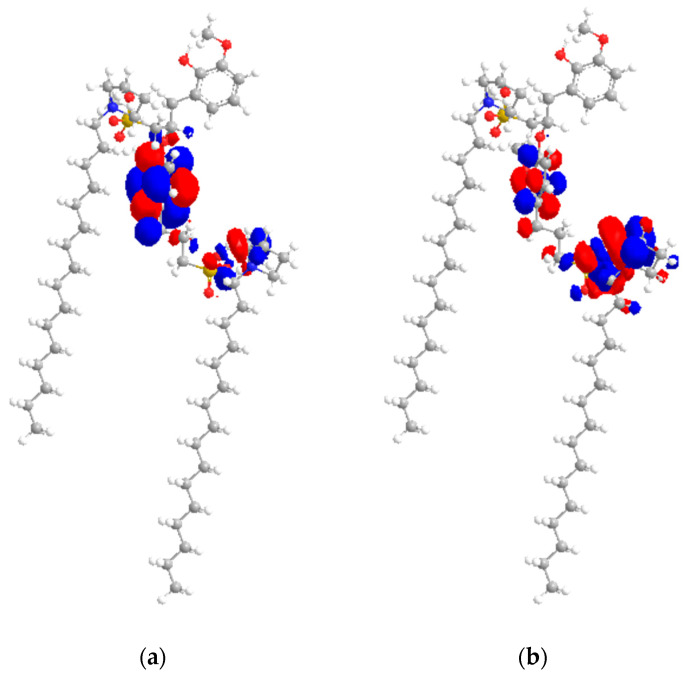
(**a**) HOMO location of [C_16_C_1_Pip]_2_[LS]; (**b**) LUMO location of [C_16_C_1_Pip]_2_[LS]. Yellow, blue, grey, white and red respectively, represented sulfur, nitrogen, carbon, hydrogen and oxygen.

**Figure 14 molecules-28-03390-f014:**
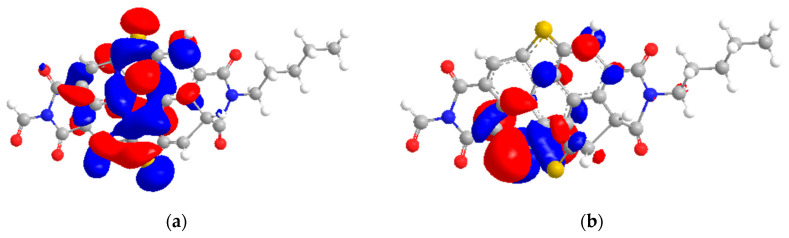
(**a**) HOMO location of asphaltene model; (**b**) LUMO location of asphaltene model [22,39]. Yellow, blue, grey, white and red respectively, represented sulfur, nitrogen, carbon, hydrogen and oxygen.

**Figure 15 molecules-28-03390-f015:**
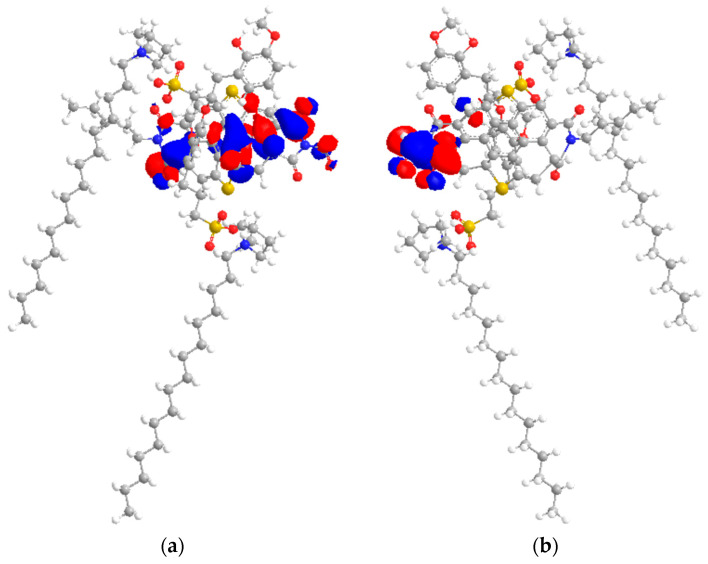
(**a**) HOMO location of [C_16_C_1_Pip]_2_[LS] + asphaltene model. Yellow, blue, grey, white and red respectively, represented sulfur, nitrogen, carbon, hydrogen and oxygen. (**b**) LUMO location of [C_16_C_1_Pip]_2_[LS] + asphaltene model.

**Figure 16 molecules-28-03390-f016:**
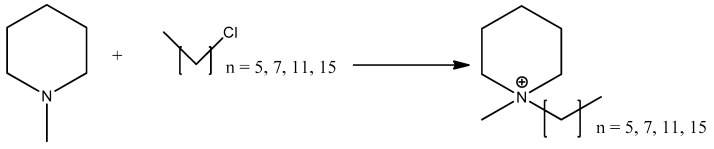
Quaternization of *n*-methyl-piperidine.

**Figure 17 molecules-28-03390-f017:**
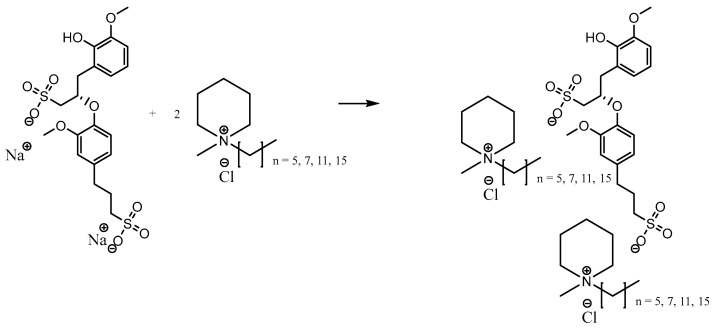
Metathesis of lignosulfonate-based ionic liquids.

**Table 1 molecules-28-03390-t001:** ILs used for petroleum asphaltene studies.

ILs	Findings	Reference
1-butyl-3-methyl-imidazolium hexafluorophosphate, [BMIM][PF_6_]	The IL possessed a higher affinity towards asphaltene to disperse the particles in non-polar media.	[28]
1,3 diheptyl-2-hydroxyphenyl-imidazolium asphaltene carboxylate, AHIL	The hydrophobic groups in the cation can increase the dispersion of asphaltenes in *n*-alkanes.	[29]
1-butyl-3-methyl-imidazolium bromide, [BMIM][Br]	A larger ionic radius exhibits better steric stabilization of IL–asphaltene complexes.	[30]
1-methyl-1-hexyl-piperidinium, [C_6_C_1_Pip]^+^, 1-methyl-1-hexyl-pyrrolidinium, [C_6_C_1_Pyr]^+^, 1-hexyl-quinolinium, [C_6_q]^+^, beryllium tetrafluoride [BF_4_]^−^ phenyl benzoate [PhBenz]^-^, benzoate [Benz]^-^	A high non-polarity cation can enhance the charge-sharing interaction with poly-aromatic molecules. An anion with high hydrophobicity can enhance the dispersing strength of asphaltene molecules.	[31]
Methyl-triooctyl-ammonium dodecyl sulfate [MTOA][DDS]	The amphiphilic property of IL provides interactions between polar and non-polar sites of asphaltene molecules.	[12]

**Table 2 molecules-28-03390-t002:** Application of lignosulfonate.

Lignosulfonate-Based	Studies	Reference
Calcium lignosulfonate	Effect of mechanical properties on treated loess	[41]
Lignosulfonate derivative of metal-sulfide catalyst	Efficiency of the hydrogenolysis process	[42]
Lignosulfonate reagent	Drilling fluid	[43]
Sodium lignosulfonate	Flotation separation of chalcopyrite and galena	[44]
Sodium lignosulfonate	Corrosion inhibition of Q235 steel in concrete pore solutions.	[45]

**Table 3 molecules-28-03390-t003:** FTIR band assignments.

Assignments	Wavenumber (cm^−1^)			Reference
	[Na]_2_[LS]	[RC_1_Pip][Cl]	[RC_1_Pip]_2_[LS]	
O-H stretch	3250	-	3250	[48]
N-H stretch	-	3250	3250	[49]
C-H stretch	2930	2930	2930	[50]
Unsaturated C=C stretch	1560		1560	[51]
C-H bend in aromatic	1460		1460	[52]
C-H bend in aliphatic		1460	1460	[53]
CH_3_ bend	1350	1350	1350	[54]
S=O stretch	1110		1110	[55]
C-O stretch of methoxy	1050		1050	

**Table 4 molecules-28-03390-t004:** Degradation temperature of lignosulfonate-based ILs.

Lignosulfonate-Based ILs	Onset Degradation Temperature (°C)	Degradation Temperature Range (°C)		
		First Stage	Second Stage	Third Stage
[C_6_C_1_Pip]_2_[LS]	118.24	118.24–207.21	207.21–305.49	453.78–800
[C_8_C1Pip]_2_[LS]	160.70	160.70–235.04	235.04–333.30	375.20–800
[C_12_C_1_Pip]_2_[LS]	204.87	204.87–254.67	254.67–375.20	333.30–800
[C_16_C_1_Pip]_2_[LS]	208.83	208.83–273.19	273.19–453.78	305.49–800

**Table 5 molecules-28-03390-t005:** Parameters of the kinetic model.

Ionic Liquid	Initial Concentration (mg/L)	Linear Pseudo-First-Order		Linear Pseudo-Second Order	
		K_1_	R^2^	K_2_	R^2^
[C_16_C_1_Pip]_2_[LS]	100	0.030	0.8652	0.035	0.9577

**Table 6 molecules-28-03390-t006:** HOMO (E_H_) and LUMO (E_L_) energies for [C_16_C_1_Pip]_2_[LS], asphaltene and [C_16_C_1_Pip]_2_[LS] with asphaltene.

Molecules	E_H_ (eV)	E_L_ (eV)	Orbital Energy Gap, E_G_ (eV)
[C_16_C_1_Pip]_2_[LS]	1.295	1.550	0.255
Asphaltene	−7.410	−6.131	1.279
[C_16_C_1_Pip]_2_[LS] + asphaltene	−2.200	−2.111	0.089

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
