# Peer review of "Lignosulfonate-Based Ionic Liquids as Asphaltene Dispersants"

_molecules, 2023, doi:10.3390/molecules28083390_

Round 1
Reviewer 1 Report
Four ionic liquids based on differently substituted methy-alkyl-piperidinium cations and the lignosulfonate anion are prepared in the present work. The use of these ionic liquids as asphaltene disperdants is investigated and the results obtained are explained in terms of HOMO-LUMO studies. The work is of interest for the readers of the journal. However, some parts need to be improved.
Table 1 should be revised. Indeed, reference 24 by the same authors reports few more ionic liquids used in asphaltene studies. Also, it would be of interest to compare the results obtained (when possible) to literature data, highlighting the role of the cation and of the anion in the application of interest.
Relevant literature on ionic liquids properties and applications should be added. For instance about ionic liquid thermal stability: https://doi.org/10.1021/ie5009597, for the negligible volatility: https://doi.org/10.1016/j.molliq.2021.115892
Biomass transformation and valorization: https://doi.org/10.1039/C9NJ00191C, https://doi.org/10.1016/j.cej.2022.136733
About the ILs synthesis, section 4.2.2, the authors should specify how the NaCl formed by the reaction is removed; why the same amount of chloride ionic liquid is used for different alkyl side chains (it sounds not correct)? Yields for all ionic liquids should be provided.
Concerning the thermal stability, section 2.3., in which way the authors can be sure that small molecules (functional groups?) decomposed in the first degradation stage? At least literature support should be provided for all degradation stage descriptions.
In the conclusions, line 558, it is stated "[RC1Pip]2[LS] has thermal stability at temperatures range of 118-800 °C." This statement is not correct as the first degradation step occurs below 270 °C for all ILs.
The English should be improved throughout the manuscript. Some verbs are missing and some parts are difficult to read. Just few examples are highlighted: Line 45 (affects not effects), line 67, lines 109-110, lines 254-256, lines 259-261, lines 268-270, lines 278-280, lines 294-296, line 376 (nucleophilicity and electrophilicity?), line 386,
Therefore, I recommend major revision before considering this work for publication.
Reviewer 2 Report
Overall
The manuscript is about asphaltene dispersion using lignosulfonate based ionic liquid.
Comments:
Abstract
-author should mention The ILs depicted high thermal stability because of the presence of long side 21 alkyl chain and pyrrolidinium cation from TGA analysis.
Introduction
- Line 48: Conventionally, the usage of a variety of organic solvents with different solubility parameters can enhance stability of asphaltene in oil. Oil is referred to model oil or crude oil. The sentence is like contradict with the aim of the work.
- Table 1: what is the anion of 1-methyl-1-hexyl-piperi-dinium, [C6C1Pip] +? ref 24
- Line 92: dianions anion?
- Line 109: The IL with the best dispersion IL?
Result discussion
- Table 3: Ref 34 assign for all assignments?
- Thermal gravimetric analysis (TGA): how about result for [C6C1Pip][Cl], [C8C1Pip][Cl], [C12C1Pip][Cl], [C16C1Pip][Cl]?
- Asphaltene dispersion study: why Cl-anion based IL have not been studied? Authors should provide/compare the result while using Cl-anion based IL for dispersion of asphaltene.
- Asphaltene aggregation study and kinetic study: why only [C16C1Pip][LS] is been studied? Authors should provide justification.
- HOMO-LUMO study: line 362 - [C16C1Pyrr]2[LS] different term is used.
Methods
- The methods are confusing. Line 437: why the product is [C6C1Pip]2[LS], shouldn’t it be n-metyl piperidinium chloride [RC1Pip][Cl]? Authors should elaborate how the ILs with 10000-50000 ppm are prepared.
- Line 502: [C16C1Pyrr]2[LS] different term is used.
- Equation (2) and (3) are not fully detailing the symbol.
Round 2
Reviewer 1 Report
The authors responded to this reviewer suggestions.
However, there are still some parts that need to be amended.
In the TGA section, lines 220-223: methoxy, hydroxyl etc are not small molecules. It would probably sound more correct to state: "Labile functional groups such as...are likely decomposed in this stage."
Line 467: "n-methyl-piperidine" should be "N-methyl-piperidine"
Furthermore, English still needs to be improved.
Author Response
Response to Reviewer 1
Point 1: In the TGA section, lines 220-223: methoxy, hydroxyl etc are not small molecules. It would probably sound more correct to state: "Labile functional groups such as...are likely decomposed in this stage."
Response 1: Track changes from line 220-223 to 220-222. Referring line 220-222, the sentence has been corrected as suggested.
Point 2: Line 467: “n-methyl-piperidine” should be “N-methyl-piperidine”
Response 2: Referring line 467, the word has been corrected.
Point 3: English still needs to be improved.
Response 3: The English has been proofread by a native English speaker